# Blood Levels of the *SMOC1* Hepatokine Are Not Causally Linked with Type 2 Diabetes: A Bidirectional Mendelian Randomization Study

**DOI:** 10.3390/nu13124208

**Published:** 2021-11-24

**Authors:** Nooshin Ghodsian, Eloi Gagnon, Jérôme Bourgault, Émilie Gobeil, Hasanga D. Manikpurage, Nicolas Perrot, Arnaud Girard, Patricia L. Mitchell, Benoit J. Arsenault

**Affiliations:** 1Centre de Recherche de l’Institut Universitaire de Cardiologie et de Pneumologie de Québec, Québec, QC G1V 4G5, Canada; nooshin.ghodsian.1@ulaval.ca (N.G.); eloi.gagnon.1@ulaval.ca (E.G.); jerome.bourgault@criucpq.ulaval.ca (J.B.); emilie.gobeil.2@ulaval.ca (É.G.); hasanga.manik-purage.1@ulaval.ca (H.D.M.); nicolas.perrot@criucpq.ulaval.ca (N.P.); arnaud.girard.1@ulaval.ca (A.G.); Patricia.Mitchell@criucpq.ulaval.ca (P.L.M.); 2Department of Medicine, Faculty of Medicine, Université Laval, Québec, QC G1V 0A6, Canada

**Keywords:** *SMOC1*, type 2 diabetes, Mendelian randomization, hepatokine

## Abstract

Hepatokines are liver-derived proteins that may influence metabolic pathways such as insulin sensitivity. Recently, Sparc-related modular calcium-binding protein 1 (SMOC1) was identified as glucose-responsive hepatokine that is dysregulated in the setting of non-alcoholic fatty liver disease (NAFLD). While *SMOC1* may influence glucose-insulin homeostasis in rodents, it is unknown if *SMOC1* is influenced by NAFLD in humans. It is also unknown if *SMOC1* is causally associated with metabolic and disease traits in humans. Therefore, we aimed to determine the effect of NAFLD on *SMOC1* gene expression in the liver and aimed to explore the potential causal associations of *SMOC1* levels with NAFLD, T2D, and glycemic traits in humans. Using an RNA sequencing dataset from a cohort of 216 patients with NAFLD, we assessed *SMOC1* expression levels across the NAFLD spectrum. We performed a series of bidirectional inverse-variance weighted Mendelian randomization (MR) analyses on blood *SMOC1* levels using two sources of genome-wide association studies (GWAS) (Fenland study, n = 10,708 and INTERVAL study, n = 3301). We utilized GWAS summary statistics for NAFLD in 8434 cases and 770,180 controls, as well as publicly available GWAS for type 2 diabetes (T2D), body mass index (BMI), waist-to-hip ratio (WHR), fasting blood insulin (FBI), fasting blood glucose (FBG), homeostatic Model Assessment of Insulin Resistance (HOMA-B and HOMA-IR), and hemoglobin A1c (HbA1C). We found that *SMOC1* expression showed no significant differences across NAFLD stages. We also identified that the top single-nucleotide polymorphism associated with blood *SMOC1* levels, was associated with *SMOC1* gene expression in the liver, but not in other tissues. Using MR, we did not find any evidence that genetically predicted NAFLD, T2D, and glycemic traits influenced *SMOC1* levels. We also did not find evidence that blood *SMOC1* levels were causally associated with T2D, NAFLD, and glycemic traits. In conclusion, the hepatokine *SMOC1* does not appear to be modulated by the presence of NAFLD and may not regulate glucose-insulin homeostasis in humans. Results of this study suggest that blood factors regulating metabolism in rodents may not always translate to human biology.

## 1. Introduction

Hepatokines are liver-secreted proteins that influence a wide range of biological processes such as lipoprotein-lipid metabolism, glucose-insulin homeostasis, inflammation, coagulation, fibrinolysis, etc. [1,2]. Therefore, hepatokines may be involved in several diseases such as non-alcoholic fatty liver disease (NAFLD), type 2 diabetes (T2D), and atherosclerotic cardiovascular diseases [3,4]. Hepatic steatosis may also induce changes in hepatokine secretion patterns and the promotion of insulin resistance [2]. A recent study indicated that the mouse liver could be responsible for the secretion of up to 538 proteins, of which 71 were dysregulated in the setting of NAFLD [5]. Sparc-related modular calcium-binding protein 1 (SMOC1) was identified as one of those hepatokines involved in glucose-insulin metabolism in primary murine hepatocytes [4]. *SMOC1* is highly expressed and secreted by the liver and regulates glucose homeostasis [6].

In mice, *SMOC1* infusions and adenoviral-mediated overexpression of *SMOC1* in the liver, significantly improved glycemic control and histological features of NAFLD, suggesting that *SMOC1* might be a potential therapeutic agent for the treatment of T2D [4,7]. Although lower *SMOC1* levels in the blood have been demonstrated in the setting of insulin resistance and obesity in humans, the causal role of *SMOC1* on the etiology of T2D and obesity in humans has not been investigated. It is therefore unknown if *SMOC1* represents a therapeutic target for the management of T2D in humans. It also remains to be determined if NAFLD in humans is associated with altered liver gene expression of *SMOC1* or alterations in the concentrations of *SMOC1* in the blood.

Mendelian randomization (MR) is a causal inference method relying on the random allocation of alleles at conception to estimate causal effects on outcomes. MR is an increasingly recognized method to determine whether genetically regulated exposures (such as blood factors) are causally linked with outcomes (such as human diseases and metabolic traits). MR has been used to determine the association between hepatokine levels and human diseases and traits. For instance, Thakkinstian et al. showed a causal association between circulating fetuin-A and BMI [8]. Other studies revealed associations between serum testosterone, sex hormone-binding globulin and several disease traits such as T2D [9,10,11].

Here, we aimed to determine whether the effect of *SMOC1* on glycemic traits in mice could be extended to humans. First, we investigated whether liver *SMOC1* expression was altered in the setting of human NAFLD. Second, using bidirectional MR), we sought to determine whether blood levels of *SMOC1* were causally influenced by NAFLD, obesity or glycemic traits and if *SMOC1* could be causally implicated in the etiology of metabolic traits and metabolic diseases such as NAFLD and T2D.

## 2. Methods

### 2.1. Observational Analysis

We first evaluated gene expression levels of *SMOC1* across different stages of NAFLD using *SMOC1* RNA expression from a cohort of 216 patients (206 NAFLD cases and 10 healthy controls) across the full histologic range of NAFLD [12]. In order to provide an accurate estimation of gene expression, we normalized the raw data from RNA sequencing with the trimmed mean of M values (TMM) method [13] and obtained gene weights using limma’s voom methodology [14]. We then performed one-way ANOVA tests and Tukey multiple comparisons of means (95% family-wise confidence level) to evaluate differences in *SMOC1* gene expression levels between two or more NAFLD groups. We compared control, NAFL, and the spectrum of NASH classifications including NASH-F0-F1 (no fibrosis), NASH-F2 (moderate fibrosis), NASH-F3 (advanced fibrosis) and NASH-F4 (severe fibrosis/cirrhosis).

### 2.2. Study Populations Included in the Mendelian Randomization Analyses

We obtained information from publicly accessible GWAS summary statistics of European ancestry with no sample overlap between exposures and outcomes. *SMOC1*: We extracted the effect of single-nucleotide polymorphisms (SNPs) in two GWAS of blood *SMOC1* levels measured in 3301 individuals of the INTERVAL study [15], and 10,708 individuals from the Fenland cohort [16]. In the INTERVAL study, Sun et al. performed genome-wide genotyping of 10.6 million imputed autosomal variants against levels of 2994 plasma proteins in 3301 individuals of European descent. The relative concentrations of 3622 plasma proteins or protein complexes were assayed using 4034 modified aptamers (SomaSCAN). In the Fenland cohort, Pietzner et al. also integrated large-scale genomic and aptamer-based plasma proteomic data from a population-based study of 10,708 individuals to characterize the genetic architecture of 179 host proteins. They characterized protein quantitative trait loci (pQTLs) in close proximity to protein-encoding genes (±500 Kb window around the gene body), cis-pQTLs. NAFLD: GWAS summary statistics for NAFLD were obtained from four cohorts including the Electronic Medical Records and Genomics (eMERGE) network, the UK Biobank, Estonian Biobank, and FinnGen with 8434 NAFLD cases and 770,180 controls [17]. T2D: Data were extracted from combined genome-wide association data of 32 studies, including 74,124 T2D cases and 824,006 controls of European ancestry [18]. Body mass index (BMI) and waist-to-hip ratio (WHR): Summary statistics were obtained from the Genetic Investigation of Anthropometric Traits (GIANT) consortium. Summary statistics for BMI were obtained from a meta-analysis of up to 125 GWAS for 339,224 European individuals [19]. Summary statistics for WHR were obtained from a meta-analysis of 210,088 individuals [20]. Glycemic traits: Summary statistics were obtained from the Meta-Analyses of Glucose and Insulin-related traits consortium (MAGIC). Summary statistics for fasting insulin (FI) and fasting glucose (FG) were obtained in 51,750 and 58,074 non-diabetic individuals respectively [21]. Summary statistics for Homeostatic Model Assessment of Insulin Resistance (HOMA-B and HOMA-IR) were obtained from 36,466 for HOMA-B and 37,037 for HOMA-IR individuals [22]. Summary statistics for hemoglobin A1c (HbA1C) were obtained from 46,368 individuals [23]. Units for continuous traits are in standard deviation (SD) and for dichotomous traits such as disease status in log (OR).

### 2.3. Mendelian Randomization Analyses

We applied MR to assess the causal effect of NAFLD, T2D, obesity and glycemic traits on blood *SMOC1* levels. We selected independent (r^2^ < 0.001) genome-wide significant SNPs (*p* < 5 × 10^8^) associated with these traits. We used a *p*-value threshold of 5 × 10^−6^ for HOMA-IR because there was no genome-wide significant instrument. We performed inverse variance weighted (IVW) analysis as primary MR analysis. For an association that reached nominal significance, we performed 6 different robust MR methods: the MR-Robust Adjusted Profile Score (MR-RAPS) [24], the contamination mixture [25], the weighted median, the weighted mode, the MR-Egger [26] and the MR-PRESSO [27] approaches. Consistent estimates across these methods provide further confirmation about the nature of the causal links. To quantify instrument strength, we used the F-statistic [28]. To quantify the variance explained we used the R^2^ [29] and to quantify estimates heterogeneity we used Cochran’s Q statistic [30].

We applied three types of MR analysis to assess the causal effect of liver *SMOC1* expression on glycemic traits. First, we used cis-MR analysis with the sentinel SNP (the one with the strongest effect on *SMOC1* levels) to calculate the Wald ratio. We selected independent (r^2^ < 0.001) genome-wide significant SNPs (*p* < 5 × 10^8^) in cis, that is 500 Kb downstream and 500 Kb upstream, of the *SMOC1* gene and performed MR analysis. Cis-acting pQTLs (close to *SMOC1*) are more specific instruments, suppressing or upregulating the expression of a gene, whereas trans pQTLs (distal to the gene) may operate via more complex mechanisms and are therefore more likely to be pleiotropic. Second, we used pan (cis + trans) MR analysis. We selected independent (r^2^ < 0.001) genome-wide significant SNPs (*p* < 5 × 10^8^) from all regions of the genome and performed MR analysis. The inclusion of trans-acting pQTL increased the number of genetic instruments, thereby increasing the power to detect association. This also allowed the use of robust analyses and the quantification of heterogeneity (Cochran’s Q) to assess the validity of the MR assumptions. The inclusion of trans pQTL, however, can introduce pleiotropy. Third, we performed multi-cis MR analyses. We included as genetic instruments all cis-acting SNP moderately independently (LD clumping r^2^ < 0.6) associated at *p* < 5 × 10^8^. We used a generalized inverse-variance weighted (IVW) model that take into account LD between SNPs. LD correlation matrix were obtained from the 1000 Genomes European ancestry reference samples [31]. This method can result in more precise estimates [32]. Altogether, the use of different MR analyses with different strengths and weaknesses increases the robustness of the causal finding.

### 2.4. Data and Code Availability

All GWAS summary statistics used in this paper are publicly available. The code used to generate these results is uploaded on GitHub: https://github.com/LaboArsenault (accessed on 25 august 2021).

## 3. Results

### 3.1. Association of Liver SMOC1 Expression with Liver Disease Progression

Differences in *SMOC1* gene expression levels across stages of steatohepatitis (control, NAFL, NASH-F0-F1, NASH-F2, NASH-F3 and NASH-F4) are presented in Figure 1. One-way ANOVA revealed that there was no significant difference in *SMOC1* RNA expression between the groups. This observation was further confirmed by Tukey’s Honestly Significant Difference (Tukey’s HSD) post-hoc test for pairwise comparisons. Liver *SMOC1* gene expression levels are therefore not associated with the progression of fatty liver disease.

### 3.2. Effect of Metabolic and Disease-Related Traits on SMOC1 Levels

The effect of metabolic diseases and glycemic traits on *SMOC1* are presented in Figure 2. Only the association of WHR with *SMOC1* in the Fenland dataset reached statistical significance, although the same association in the INTERVAL dataset did not (Figure 2 panel A). Specifically, each standard deviation increases in WHR decreased *SMOC1* blood levels by −0.23 (95% CI = −0.39, −0.066, *p*-value = 6 × 10^−3^) standard deviation. Cochran’s Q test indicated no significant heterogeneity across estimates (Q = 24.67, *p*-value = 0.92). Robust MR analysis indicated low pleiotropy as most (4/6) robust MR analyses reached nominal significance (Figure 2 panel B). These results support that waist circumference might be linked with lower *SMOC1* levels, although this result did not replicate in another dataset. Altogether, we found little evidence for a causal role of NAFLD, T2D, obesity and glycemic traits in the regulation of *SMOC1* blood levels.

### 3.3. Effect of Blood SMOC1 Levels on Metabolic and Disease-Related Traits

The effect of blood *SMOC1* levels on glycemic traits are presented in Figure 3. First, the top cis-acting SNP from both INTERVAL and Fenland (rs1958078) was strong (R^2^ = 0.05, F-statistic = 540 for Fenland study; R^2^ = 0.02, F-statistic = 55 for INTERVAL study). Power was sufficient to detect small effects (power for Fenland study: mean = 0.95, SD = 0.14; power for INTERVAL study: mean = 0.80, SD = 0.23) to detect a 0.1 effect size (beta). No significant associations between *SMOC1* and disease-related and metabolic traits were found. Second, for the pan analysis, genetic instrument selection resulted in six SNPs for Fenland study and one SNP for INTERVAL study. Therefore, the INTERVAL study did not have enough instrument to carry this analysis. The use of additional genetic variants increased the exposure’s variance explained (R^2^ = 0.078 for Fenland study). Power using this method was higher (mean = 0.97, SD = 0.08) consistent with the use of more genetic instruments. Despite high power, IVW and robust MR methods for all exposure-outcome associations were non-significant. Third, for multi-cis analysis, genetic instrument selection resulted in 97 SNPs for the Fenland study and 7 SNPs for the INTERVAL study. Theoretically, several cis-acting variants can independently affect protein expression. We modelled this biological phenomenon by correcting for the LD matrix, which can result in a more precise estimate than the sole use of cis-acting sentinel SNP in QTL analysis [32]. The mean standard error across all evaluated associations for the multi-cis method was lower (0.012) compared to the sentinel SNP method (0.18). Despite increased precision, no association was significant. Altogether, we found no evidence for a causal association between genetically predicted blood *SMOC1* levels and NAFLD, T2D, obesity as well as glycemic traits across all methods.

### 3.4. Causal Effects of SMOC1 Levels across the Human Phenome

We searched the top cis-acting SNP from both INTERVAL and Fenland studies (rs1958078) in the Phenoscanner [33,34] to discover new traits that may be associated with genetically predicted blood *SMOC1* levels. First, searching the eQTL database, rs1958078 was only and specifically associated at genome-wide significance with *SMOC1* gene level in the liver (*p* = 2 × 10^−22^). Therefore, rs1958078 is a specific genetic instrument for *SMOC1* gene expression in the liver. This provides additional evidence that changing *SMOC1* levels in the liver might not result in improved glycemic control in humans. Second, searching the pQTL database, no other protein was associated with the rs1958078 SNP or its proxies (R^2^ > 0.8). This provides further evidence that the genetic instrument is specific to *SMOC1* and unlikely to be pleiotropic. Third, searching the GWAS database, we found that rs1958075 was associated with mean corpuscular hemoglobin (i.e., the average quantity of hemoglobin present in red blood cells) and red cell distribution width (i.e., the variation in red blood cell volume and size). Using cis-MR analysis, the level of *SMOC1* protein increased mean corpuscular hemoglobin and *SMOC1* level decreased red cell distribution width consistently in both Fenland and INTERVAL’s studies. This indicates that elevating *SMOC1* plasma levels may be linked with blood-cell health outcomes. Altogether, our search in the Phenoscanner confirmed the validity of our genetic instrument and that genetically predicted levels of the *SMOC1* hepatokine do not appear to influence metabolic traits but that could be linked to other blood cells-related traits.

## 4. Discussion

To better understand the pathogenesis of metabolic diseases and to identify safe and effective prevention and treatment strategies, it is crucial to identify factors causally related to disease processes. Montgomery et al. recently proposed the hepatokine *SMOC1* as a potential therapeutic target for glycemic control based on their finding that *SMOC1* infusions and adenoviral-mediated hepatic *SMOC1* overexpression exerted a durable effect on glycemic control in mice [4]. SMOC1-increasing strategies also lead to improvements in hepatic insulin sensitivity in pre-clinical models. The authors concluded that *SMOC1* could represent a new therapeutic target that might be efficacious for treating T2D by improving glycemic control. In people with insulin resistance and obesity, *SMOC1* blood levels were statistically significantly lower than in people without these diseases. A recent study indicated that *SMOC1* is a thrombin-activating protein that makes a significant contribution to the pathophysiological changes in platelet function associated with type 2 diabetes [35]. *SMOC1* levels are correlated with the product of glucose and insulin measured during an oral glucose tolerance test [4]. These associations, however, do not provide evidence for a causal role of *SMOC1* in glucose-insulin homeostasis in humans.

The genomics revolution offers countless new opportunities to perform MR studies aimed at establishing causal relationships between a wide range of exposures and outcomes. Here, we performed bidirectional MR analyses to explore the causal associations of blood *SMOC1* levels with T2D, NAFLD, and glycemic traits. We found no evidence that NAFLD, T2D, and glycemic traits causally influenced *SMOC1* levels. We also did not find evidence that genetically predicted blood *SMOC1* levels were associated with T2D, NAFLD, and glycemic traits.

In comparison to observational analyses, the use of MR study design reduces the risk of bias from reverse causality and confounding factors. The large number of genetic variants with a strong effect on *SMOC1* levels and metabolic traits and diseases under investigation is a significant strength of our study. We also found no differences in *SMOC1* gene expression levels across stages of NAFLD. However, it should be kept in mind that the number of controls without NAFLD was rather limited (n = 10).

In conclusion, these results contrast to those previously reported in mouse models and suggest that the hepatokine *SMOC1* might not have a causal role in the pathophysiology of human metabolic traits and diseases. *SMOC1* does not appear to represent an effective target to improve glycemic control in humans, although future randomized controlled trials in humans will be required to confirm this. However, these results suggest that blood factors regulating metabolism in rodents may not always translate to human biology.

## Figures and Tables

**Figure 1 nutrients-13-04208-f001:**
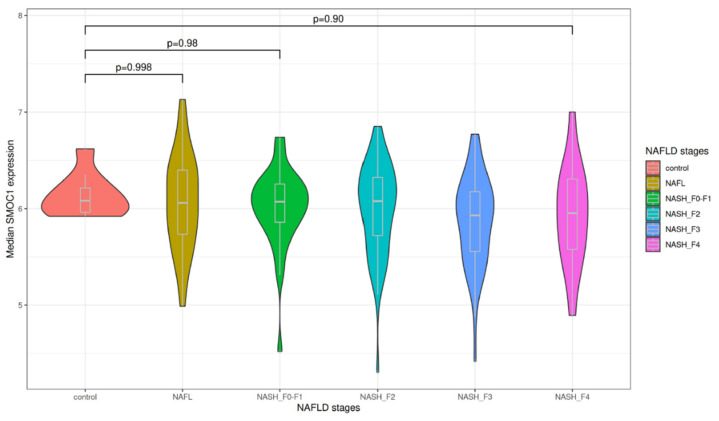
*SMOC1* gene expression levels across non-alcoholic fatty liver disease (NAFLD) stages. The violin plot indicates the mean level of *SMOC1* RNA expression across stages of steatohepatitis. This data from one-way ANOVA shows no significant difference of *SMOC1* RNA expression between any groups (control, NAFL, NASH-F0-F1, NASH-F2, NASH-F3, and NASH-F4).

**Figure 2 nutrients-13-04208-f002:**
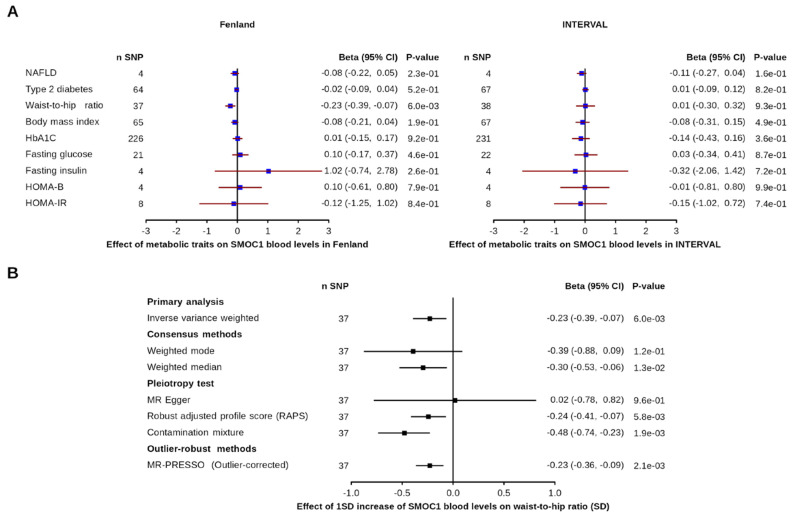
Impact of genetically predicted metabolic traits and diseases on blood *SMOC1* levels. Forest plot of Mendelian randomization estimated the effect of T2D, NAFLD, obesity and glycemic traits on blood *SMOC1* levels (panel **A**) and the effect of the waist-to-hip ratio on *SMOC1* levels across multiple MR methods (panel **B**).

**Figure 3 nutrients-13-04208-f003:**
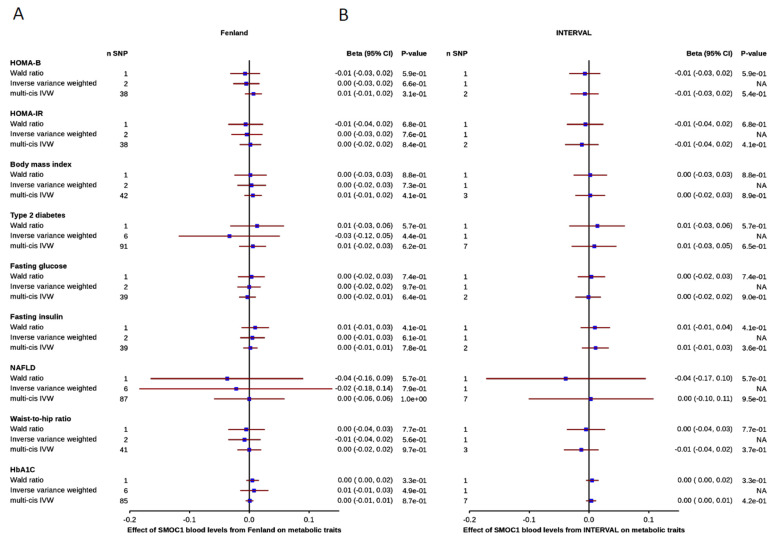
Impact of genetically predicted *SMOC1* levels on metabolic traits and diseases. Effect sizes and 95% confidence intervals for *SMOC1* SNPs derived from two GWAS studies (panel **A**) Fenland and (panel **B**) INTERVAL study. The blue cubes at each plot represent the Wald ratio estimate, inverse variance weighted (IVW) and multi cis IVW with 95% confidence interval.

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
