# Peer review of "Blood Levels of the SMOC1 Hepatokine Are Not Causally Linked with Type 2 Diabetes: A Bidirectional Mendelian Randomization Study"

_nutrients, 2021, doi:10.3390/nu13124208_

Round 1

Reviewer 1 Report

A brief introduction of other studies using the MR method for the association between hepatokines and human diseases / metabolic traits would be appreciated.

It would be of great interest for the observational analysis to add a table with the most relevant clinical characteristics of the 10 healthy controls and the 206 NAFLD cases. Moreover, due to the reduced number of controls, a comparison of liver SMOC1 RNA expression between controls and a subset of 1:1 or 1:2 age-, sex- and BMI-matched NAFLD cases would be really interesting to rule out their role as confounding factors.

Previous reports on the upregulation of SMOC1 in platelets from individuals with type 2 diabetes should be cited in the Discussion.

In line 65, the abbreviation for Mendelian randomization (MR) is introduced for the second time. The abbreviation for Hemoglobin A1c (HbA1C) has to be introduced in the main text.

Methodology citations such as Robinson and Oshlack 2010 (line 76), Ritchie et al. 2015 (line 77), Soranzo et al. 2010 (line 111), Burgess, Thompson, 124 and CRP CHD Genetics Collaboration 2011 (line 124), Burgess, Bowden, et al. 2017 (line 126) and Burgess, Zuber, et al. 2017 (145) should be added to the References Section.

Line 147 different text format. In line 194, surplus dot "(R2 = 0.078. for Fenland study)".

Author Response

Dear Reviewer

We are so grateful to have this opportunity to send you the revised version of our recent paper entitled “Blood levels of the SMOC1 hepatokine are not causally linked with type 2 diabetes: A bidirectional Mendelian randomization study” for publication in Nutrients. Thanks to both reviewers for your comments on our paper. We have been able to incorporate changes to reflect most of the your suggestions. All changes within the manuscript were conducted using track change. Here is a point-by-point response to the reviewers' comments and concerns.

Comment 1: A brief introduction of other studies using the MR method for the association between Hepatokines and human diseases / metabolic traits would be appreciated.

Response: Thank you for your comment. We added three references which performed MR method to identify the association of hepatokines with human traits. Line: 140-144.

Comment 2: It would be of great interest for the observational analysis to add a table with the most relevant clinical characteristics of the 10 healthy controls and the 206 NAFLD Moreover, due to the reduced number of controls, a comparison of liver SMOC1 RNA expression between controls and a subset of 1:1 or 1:2 age-, sex- and BMI-matched NAFLD cases would be really interesting to rule out their role as confounding factors.

Response: We agree with the reviewer. We added that the small number of healthy controls is a limitation to these findings in the discussion section. Since these results were null, we find it very unlikely that confounders could explain this relationship. Unfortunately, we don’t have much clinical data on these individuals. Line 519-521.

Comment 3: Previous reports on the upregulation of SMOC1 in platelets from individuals with type 2 diabetes should be cited in the Discussion. Recent study indicated that SMOC1 is a thrombin-activating protein that makes a significant contribution to the pathophysiological changes in platelet function associated with type 2 diabetes. 

Response: Thank you for pointing this out. We have added this reference in the discussion. Line 459-461.

Comment 4: In line 65, the abbreviation for Mendelian randomization (MR) is introduced for the second time.

Response: Thank you for this comment. We have removed the second Mendelian randomization from text.

Comment 5: The abbreviation for Hemoglobin A1c (HbA1C) has to be introduced in the main text.

Response: Response: We agree with this comment. We have accordingly corrected the text. Line 296.

Comment 6: Methodology citations such as Robinson and Oshlack 2010 (line 76), Ritchie et al. 2015 (line 77), Soranzo et al. 2010 (line 111), Burgess, Thompson, 124 and CRP CHD Genetics Collaboration 2011 (line 124), Burgess, Bowden, et al. 2017 (line 126) and Burgess, Zuber, et al. 2017 (145) should be added to the References Section.

Response: Thank you for this important comment. Missing references are added.

Comment 7: Line 147 different text format. In line 194, surplus dot "(R2 = 0.078. for Fenland study)".

Response: dot in "(R2 = 0.078. for Fenland study)" is removed. Line 403.

Reviewer 2 Report

The paper reads well. The presentation is clear, and the writing is well-structured. 

Limitations of this report include the relatively low statistical power when investigating the relevance of NAFLD for SMOC1. Hence, the conclusions may need to be toned down. 

Author Response

Dear reviewer

We are so grateful to have this opportunity to send you the revised version of our recent paper entitled “Blood levels of the SMOC1 hepatokine are not causally linked with type 2 diabetes: A bidirectional Mendelian randomization study” for publication in Nutrients. Thank you for your comment on our paper. We have been able to incorporate changes to reflect your suggestion. All changes within the manuscript were conducted using track change. 

Comment : Limitations of this report include the relatively low statistical power when investigating the relevance of NAFLD for SMOC1. Hence, the conclusions may need to be toned down. 

Response: We agree with the reviewer. We added this as a limitation to our study. Line 519-521.